# Tubeless Percutaneous Nephrolithotomy in the Barts ‘Flank-Free’ Modified Supine Position with 24-Hour Discharge: A Single-Center Experience

**DOI:** 10.3390/medicina61040748

**Published:** 2025-04-18

**Authors:** Zoltán Kiss, Gyula Drabik, Mihály Murányi, Attila Nagy, Ioannis Kartalas Goumas, Tibor Flaskó

**Affiliations:** 1Department of Urology, University of Debrecen, 4032 Debrecen, Hungary; drabik.gyula@med.unideb.hu (G.D.); muranyi.mihaly@med.unideb.hu (M.M.); flash@med.unideb.hu (T.F.); 2Department of Health Informatics, Faculty of Health Sciences, University of Debrecen, 4032 Debrecen, Hungary; attilanagy@med.unideb.hu; 3Clinical Institute Beato Matteo, 27029 Vigevano, Italy

**Keywords:** PCNL, ECIRS, supine, tubeless, Barts position, urolithiasis

## Abstract

*Background and Objectives:* To evaluate the effectiveness and outcomes of supine percutaneous nephrolithotomy (PCNL) using the Barts ‘flank-free’ position and ultrasound-guided puncture, assessing the feasibility of the tubeless technique for discharge within 24 h. *Materials and Methods:* We conducted a retrospective analysis of 208 patients across 220 renal units who underwent supine PCNL at a tertiary university hospital between May 2019 and December 2024. All procedures were performed by a single surgeon. Patient demographics, stone characteristics, and surgical outcomes were analyzed. The tubeless technique was applied in most cases, and outcomes were assessed in terms of operative time, complication rates, stone-free rates (SFRs), and length of hospital stay. *Results:* The mean operating time was 50.34 ± 30.80 min. Single-tract PCNL was performed in 94.55% of cases, with the tubeless technique used in 90% of patients. The overall complication rate was 9.55%, with no Clavien–Dindo grade IV–V complications observed. On the first postoperative day, 68.18% of patients were discharged, demonstrating 24 h discharge feasibility. SFR and complication rates aligned with existing literature. *Conclusions:* The Barts ‘flank-free’ position and ultrasound-guided puncture considerably improved surgical access and safety in supine PCNL. The tubeless technique facilitates faster recovery, making early discharge feasible, even with standard sheath sizes. Further research is warranted to validate these findings and optimize renal stone management outcomes.

## 1. Introduction

Percutaneous nephrolithotomy (PCNL) is currently the gold-standard treatment for kidney stones larger than 20 mm and can be safely performed in either the prone or supine position, each with well-documented advantages and disadvantages [1].

There are several variations of the supine position, each differing in the placement of saline bags or gel pads beneath the patient, the degree of tilt, and the positioning of the legs. Selecting the appropriate position is crucial not only for the surgeon but also for the patient. The Barts ‘flank-free’ position, using a 15-degree tilt, offers numerous advantages. It eliminates the need for support under the flank, providing more space for optimal renal access. Additionally, the kidney remains in a neutral position, reducing the likelihood of anterior displacement. This stability minimizes kidney mobility, facilitating both puncture and dilation. The nearly horizontal orientation of the percutaneous tract in this position facilitates better washout of fragments and maintains lower intrarenal pressure. It is particularly advantageous for multitract PCNL, as it provides more space by keeping the flank unobstructed, making it an ideal choice for this method. Additionally, the minimal trunk rotation allows for easier and more ergonomic retrograde access for the second surgeon, further supporting its indication for this approach [2].

Nonetheless, PCNL can still lead to several complications. According to the literature, low-grade, medium-grade, and severe complications occur in 16.4%, 3.6%, and 0.5% of cases, respectively [3]. These complications include urinary tract infections, hematuria requiring blood transfusion, urinary tract leakage, and organ injury. Our literature review identified only one study that compared complication rates among different supine positions, in which the authors concluded that the rate of severe complications is not influenced by the specific type of supine position used [4].

Traditional PCNL typically concludes with the placement of a nephrostomy tube, requiring 2–3 d of postoperative observation. In contrast, day-case PCNL uses the tubeless technique, substantially reducing hospital stay. This approach lowers healthcare costs, speeds up recovery, and reduces the burden on the healthcare system [5,6].

In our retrospective study, we reviewed our experience with 220 cases of supine PCNL using a standard sheath size. The study focused on the effectiveness and outcomes of this technique. We hypothesized that the tubeless method facilitates discharge within 24 h.

## 2. Materials and Methods

Between May 2019 and December 2024, a total of 208 patients across 220 renal units underwent supine PCNL at a tertiary university hospital. Following approval from the Regional Institutional Research Ethics Committee (IRB No. DERKEB/IKEB 6969-2024) and the acquisition of written informed consent, data were retrospectively collected. Inclusion criteria for the study were patients with single or multiple kidney stones > 20 mm, as well as those with smaller stones who had previously not responded to extracorporeal shock wave lithotripsy or had failed flexible ureterorenoscopy (fURS). In cases involving complex stones, multiple stones, or stones located in parallel calyces or the ureter, endoscopic combined intrarenal surgery (ECIRS) was employed. Exclusion criteria included individuals with bleeding disorders, pregnancy, and untreated urinary tract infections. All procedures were conducted by a single surgeon experienced in endourology. For the ECIRS technique, the retrograde approach was carried out by another surgeon who specialized in fURS.

Patient demographics—including age, sex, body mass index (BMI) and American Society of Anesthesiologists (ASA) score—were analyzed. For imaging, the patients underwent urinary tract ultrasound; kidney–ureter–bladder (KUB) X-ray; and low-dose abdominal and pelvic computed tomography (CT) in supine position to access stone size, position, and density. Stone volume was calculated using the spheric formula (V = 4/3 × π × radius^3^). The Guy’s stone score was used to predict surgical complexity.

Additionally, urine cultures and laboratory parameters (creatinine and hemoglobin levels) were also analyzed.

### 2.1. Surgical Technique

Intravenous antibiotics were administered to the patients 30 min before the start of surgery. Typically, a broad-spectrum third-generation cephalosporin, such as 2 g of ceftriaxone, was used.

Under general or spinal anesthesia, the patients were positioned in the Barts ’flank-free’ modified supine position with a 15-degree tilt, ensuring special attention to protect pressure points. The ipsilateral leg was extended, and the contralateral leg was flexed (Figure 1). In obese patients, adhesive tape was used to shift the abdomen towards the contralateral side (Figure 2). Key anatomical landmarks, including the posterior axillary line, rib line, and iliac crest, were marked on the patient (Figure 3). A 19 Ch rigid cystoscope (Urotech GmbH, Achenmühle, Germany) was inserted to identify the ipsilateral ureteral orifice, and a 5 Ch open-end ureteral catheter (Urotech GmbH, Achenmühle, Germany) was advanced into the ureter. Ultrasound (Mindray Diagnostic Ultrasound System, Consona N9, Shenzhen, China) assessed perirenal anatomy, the position of adjacent organs relative to the kidney, and target calyx selection. An ultrasound-guided, fluoroscopically adjusted puncture was performed using an 18-gauge needle (Cook Incorporated, Bloomington, IN, USA) under the posterior axillary line, advancing a 0.035 mm hydrophilic guidewire (Roadrunner^®^, Cook Incorporated, Bloomington, IN, USA) into the ureter. Dilatation was achieved using an Amplatz renal dilator set (Cook Incorporated, Bloomington, IN, USA) under fluoroscopy, followed by the placement of a 28 Ch Amplatz sheath and a 25 Ch nephroscope (Olympus Winter & IBE GmbH, Hamburg, Germany). The guidewire ensured safety throughout the procedure via the Amplatz sheath. Lithotripsy was performed using an ultrasound lithotripter (ShockPulse-SE Lithotripsy System^TM^, Cybersonics, Inc., Erie, PA, USA).

For ECIRS, a 6 Ch double-J stent (Urotech GmbH, Achenmühle, Germany) was placed for the purpose of pre-stenting two weeks before surgery. After inserting the hydrophilic guidewire, a second guidewire was advanced into the ureter. A 35 or 46 cm-long 11 Ch/13 Ch access sheath (Navigator™, Boston Scientific Corporation, Spencer, IN, USA) was placed, and an 8.4 or 7.5 Ch single-use flexible ureteroscope (Scivita Medical Technology Co., Ltd., Suzhou, China) was advanced to the renal pelvis. In ECIRS procedures, lower calyx access was preferred in all cases. For multiple small stones, the Ngage^TM^ Nitinol Stone Extractor (Cook Incorporated, Bloomington, IN, USA) repositioned stones to the renal pelvis for easy removal via the Amplatz sheath using the “pass-the-ball” technique. If this technique was not feasible owing to stone size or a narrow infundibulum, laser lithotripsy was performed using a 30 W holmium laser (Quanta System S.p.A., Samarate, Italy). A 6 Ch double-J stent was placed retrogradely at the end of the procedure. In this case, tubeless PCNL was considered. In cases of prolonged operative time or multitract access, a nephrostomy tube was also inserted. To reduce postoperative pain, a local injection of diluted bupivacaine was administered into the percutaneous tract. Additionally, the anesthesiologist administered tranexamic acid during surgery. The bladder catheter was removed on the first postoperative day, and the double-J stent was removed a week later.

### 2.2. Outcome Measures

The operative time, position and number of tracts, method of urine diversion, and the length of hospital stay were evaluated. On the first postoperative day, hemoglobin and creatinine levels were monitored, and KUB radiography was performed to verify the correct position of the double-J stent. Complications were assessed using the Clavien–Dindo classification. Early discharge was considered when the patient met the following criteria: urine without blood clots; absence of renal colic pain, nausea, or vomiting; confirmation of the double-J stent’s correct position on the control KUB X-ray; and the patient’s acceptance of discharge. One month postoperatively, KUB radiography and ultrasound assessed the stone-free rate (SFR). As CT was not used to confirm SFR, residual fragments less than 4 mm were considered stone-free for SFR assessment.

### 2.3. Statistical Analysis

The Shapiro–Wilk test was employed to assess the normality of continuous variables. For variables with a normal distribution, results are presented as means ± standard deviations, and comparisons were conducted using parametric tests (e.g., independent or paired *t*-tests as appropriate). For non-normally distributed variables, medians with interquartile ranges are reported, and non-parametric tests (e.g., the Wilcoxon signed-rank test) were used. Categorical variables are expressed as frequencies and percentages. Statistical analyses were performed using the Intercooled Stata v18.0. (StataCorp. 2023. Stata Statistical Software: Release 18. StataCorp LLC., College Station, TX, USA), with statistical significance set at *p* < 0.05.

## 3. Results

Patient demographics and stone characteristics are presented in Table 1. The mean operating time was 50.34 ± 30.80 min. In total, two hundred and seventeen (98.64%) surgeries were performed under general anesthesia, and three (1.36%) under spinal anesthesia. Single-tract PCNL was performed in 208 (94.55%) cases, whereas multitract PCNL was performed in 12 (5.45%) cases. Lower, upper, and middle calyx access was achieved in one hundred and eighty-seven (85%), fourteen (6.36%), and seven (3.18%) cases, respectively. Multitract PCNL included lower and upper tract positions in five (2.27%) cases, lower and middle in four (1.82%) cases, and middle and upper in two (0.91%) cases. Three-tract PCNL was performed on only one patient (0.45%). The tubeless technique was used in 198 patients (90%), whereas both a double-J stent and nephrostomy tube were placed in 22 patients (10%). The postoperative hemoglobin decrease was 15.31 ± 12.09 g/L, which was statistically significant (*p* < 0.05). The median change in creatinine level was −7 µmol/L [−17.5 to 1.5], which was not statistically significant. The overall complication rate was 9.55%. Clavien–Dindo grade I complications occurred in 11 patients. Hematuria was observed in 10 patients (4.55%) postoperatively and managed with prolonged catheterization and infusions. One patient (0.45%) experienced urine leakage after nephrostomy tube removal, which resolved spontaneously. Clavien–Dindo grade II complications were observed in nine patients. Fever was observed in seven (3.18%) patients who were managed with parenteral antibiotics. One patient (0.45%) required a blood transfusion and one (0.45%) developed urosepsis, which was managed with prolonged parenteral antibiotic treatment. A Clavien–Dindo grade IIIa complication was observed in one patient (0.45%), requiring readmission 2 weeks post-surgery owing to severe hematuria. CT angiography and selective angioembolization were performed because of the presence of an arteriovenous fistula. No Clavien–Dindo grade IV–V complications occurred. On the first postoperative day, 150 patients (68.18%) were discharged. All patients discharged within 24 h were tubeless and had single-tract access. Among tubeless cases (*n* = 198), the early discharge rate was 73.23% (*n* = 145).

Late discharge occurred in 70 patients (31.82%) because of the aforementioned complications, nephrostomy tube placement, and socioeconomic reasons, which occurred in 19 (8.64%), 22 (10%), and 29 (13.18%) patients, respectively. The overall SFR and SFR in each Guy’s stone score groups are presented in Figure 1.

## 4. Discussion

Prone PCNL—introduced by Fernstrom and Johansson in 1976—revolutionized the treatment of kidney stones and marked a paradigm shift in urological practice [7]. Valdivia laid the cornerstone for supine PCNL, leading to several positioning modifications of patients [8,9]. The choice between prone and supine PCNL remains debated, depending on surgeon preference or the patient characteristics.

Traditionally, the prone position was preferred because of concerns about colonic injury. However, with the widespread adoption of CT scans, the previous dogma has been challenged, as these scans have demonstrated that the risk of colonic injury is lower in the supine position. Studies have shown that the incidence of retrorenal and posterolateral colon is 15.7% in the supine position compared to 24.5% in the prone position [10]. The supine position improves airway control, reduces radiation exposure for the surgeon, lowers intrarenal pressure, aids spontaneous fragment clearance, and eliminates repositioning and redraping of the patient [11]. Additionally, it facilitates easier access to the upper pole of the lower pole tract. Supine PCNL is particularly advantageous for patients with spinal deformities, cardiovascular issues, and obesity [1]. A key advantage of supine position is its suitability for ECIRS and Simultaneous Bilateral Endoscopic Surgery [12]. ECIRS combines antegrade and retrograde approaches, optimizing visibility via dual irrigation of the collecting system. fURS provides access to all calyces, reducing the need for multiple tracts in PCNL, thereby decreasing bleeding risk. The retrograde approach is particularly beneficial for assisting with puncture and dilation, which reduces fluoroscopic exposure for both the patient and the surgical team. Additionally, ECIRS enables full ureter examination and simultaneous ureteral stone removal during the same session. Furthermore, the SFR after ECIRS is higher compared to traditional PCNL [13].

However, the supine position also has its drawbacks, including a narrower access space, kidney mobility making puncture more challenging, and difficulty accessing the upper calyx. Despite these challenges, supine PCNL has considerable benefits. Nevertheless, it is not yet a widespread practice, with the ratio of prone to supine PCNL being approximately 80% to 20% among urologists [14].

Regarding the evaluation of patients, for whom PCNL is planned, the position of the patient during noncontrast computed tomography (NCCT) is crucial, as it affects kidney and organ orientation. In this study, NCCT was performed in the supine position for all patients, aiding surgical access planning, as PCNL was also performed in the same position [15]. Among the various supine positioning options, we preferred the Barts ‘flank-free’ position. Gel pads or saline bags placed under the iliac crest and rib cage created a clear working area for nephroscope maneuverability [16]. To prevent interference between the nephroscope and the operating table, the patient was positioned as close to the edge of the table as possible. To increase safety, ultrasound-guided puncture was favored, offering several benefits. Ultrasound can verify renal and perirenal anatomy, enhance visualization of adjacent organs, distinguish between anterior and posterior calyces, reduce radiation exposure, and provide real-time imaging of the collecting system [17,18].

The optimal method of urine diversion remains an ongoing discussion. Placing a nephrostomy tube improves drainage of infected urine, enables second-look procedures, tamponades the percutaneous tract, and prevents urinoma formation. However, the tubeless technique offers faster recovery, less postoperative pain, reduced analgesic requirements, shorter hospital stays, and less urinary leakage. Nevertheless, an additional outpatient procedure is required for double-J stent removal unless strings are used [19]. Our results indicate that the tubeless technique facilitates early discharge. No significant differences in SFR and complications were found between prone and supine positions [20,21]. This suggests that the choice of position can be based on surgeon preference and the specific clinical scenario without compromising outcomes. Our SFR and complication rates were consistent with existing literature, supporting our approach [22]. However, there is a discrepancy between the low reported rate of hematuria and the data on anemia. This inconsistency may be due to the lack of data on the incidence of mild hematuria and the absence of precise calculations of intraoperative blood loss.

The limitations of our study included its single-center, retrospective design, affecting generalizability of the findings. Additionally, low-dose CT was not used to confirm SFR because of the high burden on the healthcare system in our country, limiting resource availability.

## 5. Conclusions

In conclusion, our study supports the use of the Barts ‘flank-free’ position and ultrasound-guided puncture in supine PCNL, highlighting their benefits in terms of surgical access and safety. The tubeless technique offered a viable alternative to traditional nephrostomy tube placement, promoting faster recovery. Our findings suggest that discharge within 24 h is feasible with single-tract access and the tubeless technique, even when using a standard sheath size. Continued research is essential to refine these techniques and optimize patient outcomes in the management of renal stones.

## Data Availability

The data that support the findings of this study are available upon reasonable request.

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
