# Peer review of "Tubeless Percutaneous Nephrolithotomy in the Barts ‘Flank-Free’ Modified Supine Position with 24-Hour Discharge: A Single-Center Experience"

_medicina, 2025, doi:10.3390/medicina61040748_

Round 1
Reviewer 1 Report
Comments and Suggestions for Authors
To thank the authors for an original article on the benefits of a novel technique for percutaneous nephrolithotomy. The article is well written, clear and concise.
Only a few recommendations to improve the quality of the manuscript:
1.- The introduction is somewhat short, it should be expanded with some epidemiological data on the cases in which this technique is used with clear indications, a short description of the benefits of using this technique and not others, add data on complications of the procedure.
2.-Line 48, remove ¨with¨ change to ¨of¨.
3.- In the surgical technique section, is not prophylactic antibiotic treatment added? if so, please add dosage and type of antibiotic.
4.-In the outcome measures, the term clear urine is very subjective, unclear, what do you mean by a very dilute urine, if so, describe the urinary density, otherwise modify it by urine without clots, which I imagine is what you mean by the term clear urine.
5.- In the result section, there is mention of a significant drop in Hb almost 3 points on average, and this data is not discussed in the discussion, it makes us think that it is an anemizing technique, why? In this case, in the indication protocol, it should be added that patients with Hb less than 11 should not undergo this technique.
6.- The low rate of hematuria is not consistent with the anemization data, this should be mentioned in the discussion.
7.- The sentence in lines 191-192 is not clear, rephrase it.
8.- In the paragraph on line 196, add a bibliographic citation.
Author Response
1.- The introduction is somewhat short, it should be expanded with some epidemiological data on the cases in which this technique is used with clear indications, a short description of the benefits of using this technique and not others, add data on complications of the procedure.
Thank you for your suggestion. We have expanded the introduction to provide a more comprehensive overview of this method. We have also added three new references to support these data and statements.
Lines in the revised version: 38-57
References: 2-4
There are several variations of the supine position, each differing in the placement of saline bags or gel pads beneath the patient, the degree of tilt, and the positioning of the legs. Selecting the appropriate position is crucial not only for the surgeon but also for the patient. The Barts flank-free position, using a 15-degree tilt, offers numerous advantages. It eliminates the need for support under the flank, providing more space for optimal renal access. Additionally, the kidney remains in a neutral position, reducing the likelihood of anterior displacement. This stability minimizes kidney mobility, facilitating both puncture and dilation. The nearly horizontal orientation of the percutaneous tract in this position facilitates better washout of fragments and maintains lower intrarenal pressure. It is particularly advantageous for multitract PCNL, as it provides more space by keeping the flank unobstructed, making it an ideal choice for this method. Additionally, the minimal trunk rotation allows for easier and more ergonomic retrograde access for the second surgeon, further supporting its indication for this approach [2].
Nonetheless, PCNL can still lead to several complications. According to the literature, low-grade, medium-grade, and severe complications occur in 16.4%, 3.6%, and 0.5% of cases, respectively [3]. These complications include urinary tract infections, hematuria requiring blood transfusion, urinary tract leakage, and organ injury. Our literature review identified only one study that compared complication rates among different supine positions, in which, the authors concluded that the rate of severe complications is not influenced by the specific type of supine position used [4].
2.-Line 48, remove ¨with¨ change to ¨of¨.
Thank you for your feedback, and we apologize for the grammatical error. Following your suggestion, we found that a more suitable modification would be revising ¨with¨ to ¨across.¨
Line in the revised version: 69
3.- In the surgical technique section, is not prophylactic antibiotic treatment added? if so, please add dosage and type of antibiotic.
Thank you for your valuable suggestion. As a standard practice, we always administer prophylactic antibiotics before surgery. We have revised the manuscript accordingly.
Intravenous antibiotics were administered to the patients 30 min before the start of surgery. Typically, a broad-spectrum third-generation cephalosporin, such as 2 grams of ceftriaxone, was used.
Lines in the revised version: 93-95
4.-In the outcome measures, the term clear urine is very subjective, unclear, what do you mean by a very dilute urine, if so, describe the urinary density, otherwise modify it by urine without clots, which I imagine is what you mean by the term clear urine.
Thank you for your valuable comment, which we completely agree with. The concept of clear urine is subjective and can vary significantly between observers. We consider early discharge appropriate when the urine is free of blood clots. We have revised the manuscript to reflect this.
Lines in the revised version: 144
5.- In the result section, there is mention of a significant drop in Hb almost 3 points on average, and this data is not discussed in the discussion, it makes us think that it is an anemizing technique, why? In this case, in the indication protocol, it should be added that patients with Hb less than 11 should not undergo this technique.
Thank you for your valuable comment. The postoperative hemoglobin decrease was 15.31 ± 12.09 g/L, which was statistically significant (p <0.05), but not clinically significant. In the postoperative period, all patients were hemodynamically stable. Only one patient required a blood transfusion, which is consistent with findings reported in the literature. As you rightly pointed out, in patients with preoperative anemia, we would prefer to perform a mini-PCNL rather than a standard-sheath PCNL. We do not consider this technique to be significantly anemizing. However, it is important to note that several factors can influence blood loss during PCNL, including the size of the sheath, number of access tracts, and operative time. In our study, 12 cases involved multritract access, which may have contributed to increased blood loss.
6.- The low rate of hematuria is not consistent with the anemization data, this should be mentioned in the discussion.
Thank you for your valuable comment, which we completely agree with. Blood loss can occur intraoperatively, as well as in the early and delayed postoperative periods. The literature is inconsistent regarding complications of PCNL, such as hematuria. Some authors do not consider mild, clinically insignificant, transient hematuria to be a complication. We agree with this perspective and defined hematuria as gross hematuria that requires observation, prolonged catheterization, infusions, and tranexamic acid. In our study, gross hematuria was observed in 10 patients and was managed conservatively. The inconsistency in results can be attributed to the lack of data on the rate of mild hematuria and the absence of calculated intraoperative blood loss.
Lines in the revised version: 260-263
7.- The sentence in lines 191-192 is not clear, rephrase it.
Thank you for your insightful suggestion. We have rephrased the corresponding sentence accordingly.
However, with the widespread adoption of CT scans, the previous dogma has been challenged, as these scans have demonstrated that the risk of colonic injury is lower in the supine position.
Lines in the revised version: 216-218
8.- In the paragraph on line 196, add a bibliographic citation.
Thank you for your pertinent suggestion. We have cited the corresponding reference accordingly.
Reference number: 11
Reviewer 2 Report
Comments and Suggestions for Authors
This study evaluated the role of tubeless percutaneous nephrolithotomy in the Barts flank free modified supine position in patients with > 2cm renal stones. This single center study is trying to point out the advantages and the drawbacks of this method. The choise of topic is intersting because position matters in supine PCNL and a well-chosen position like Barts flank free can improve outcomes, reduce complications and make the procedure smoother for both patient and surgeon. Furthermore not many studies in literature point out the importance of the Barts flank free supine PCNL. The only drawback of the study is that it is a single center study and the data should be validated from other studies.
Author Response
This study evaluated the role of tubeless percutaneous nephrolithotomy in the Barts flank free modified supine position in patients with > 2cm renal stones. This single center study is trying to point out the advantages and the drawbacks of this method. The choise of topic is intersting because position matters in supine PCNL and a well-chosen position like Barts flank free can improve outcomes, reduce complications and make the procedure smoother for both patient and surgeon. Furthermore not many studies in literature point out the importance of the Barts flank free supine PCNL. The only drawback of the study is that it is a single center study and the data should be validated from other studies.
Thank you for your valuable review and kind words. We agree that one of the limitations of our study is its unicentric design. Consequently, we have highlighted this limitation at the end of the Discussion section.
Line in the revised version: 264